# Serum and Prostatic Tissue Concentrations of Cefazolin, Ciprofloxacin and Fosfomycin after Prophylactic Use for Transurethral Resection of the Prostate

**DOI:** 10.3390/antibiotics12010022

**Published:** 2022-12-23

**Authors:** Annemieke Sobels, Koen J. Lentjes, Frank M. J. A. Froeling, Cees van Nieuwkoop, Erik B. Wilms

**Affiliations:** 1Department of Hospital Pharmacy, Haga Teaching Hospital, 2545 AA The Hague, The Netherlands; 2Department of Urology, Haga Teaching Hospital, 2545 AA The Hague, The Netherlands; 3Department of Urology, Lange Land Hospital, 2725 NA Zoetermeer, The Netherlands; 4Department of Internal Medicine, Haga Teaching Hospital, 2545 AA The Hague, The Netherlands; 5Laboratory of Pharmaceutical Analysis and Toxicology, The Hague Hospital Pharmacy, 2545 AB The Hague, The Netherlands

**Keywords:** TURP, prostate biopsy, antibiotics, prophylaxis, fosfomycin

## Abstract

The optimal drug of choice, its time of administration and duration of antibiotic prophylaxis in patient undergoing a TURP procedure are still matters of debate. In this study, we evaluated the concentrations of cefazolin, ciprofloxacin and fosfomycin in the human prostate in a cohort of men undergoing TURP. We compared prostate tissue concentrations to the serum concentrations and MICs of common uropathogens, to determine the appropriateness of the current presurgical prophylactic antibiotics and to gain supportive data about the suitability of fosfomycin for antibiotic prophylaxis in men undergoing urological procedures of the prostate. After a single intravenous dose of cefazoline or an oral dose of ciprofloxacin prior to TURP, concentrations in serum and prostate tissue of well above the MIC (EUCAST breakpoint) of common uropathogens (*Enterobacterales*) were reached, and both antibiotics seem potentially effective in preventing postsurgical infections. A single dose of oral and intravenous administration of fosfomycin both led to serum concentrations above the MIC for uncomplicated urinary tract infections (8 µg/mL). The MIC for other infections (32 µg/mL) was only reached after a single dose of intravenous fosfomycin. We were unable to detect fosfomycin concentrations in prostate tissue.

## 1. Introduction

Postoperative infections, such as bacteremia and prostatitis, are known complications after transurethral resection of the prostate (TURP); their incidence is 6–64%, depending on the type of urethral instrumentation used and the use of prophylactic antibiotics [1,2]. This is associated with prolonged hospital stay and increased costs. Prophylactic antibiotic use in TURP significantly decreases the incidence of bacteriuria and prostatitis, even in men with preoperative sterile urine [3]. The optimal drug of choice, its time of administration and duration of antibiotic prophylaxis depend merely on local antibiotic resistance patterns of the most common uropathogens, but it remains matter of international debate [4].

In The Netherlands, the most common regimens used for antibiotic prophylaxis in patients undergoing TURP are intravenous cefazolin, 1–2 g immediately prior to surgery, intravenous ciprofloxacin, 400 mg immediately prior to surgery or oral ciprofloxacin 500 mg 1–2 h before surgery, aiming to achieve peak serum concentrations during the TURP procedure. Whether this also reflects peak concentration in prostatic tissue is still unclear as only limited information about tissue distribution and kinetics in the prostate is available.

Infection of the prostate caused by multidrug-resistant (MDR) uropathogens is a growing problem that limits the use of fluoroquinolones and cephalosporines as prophylactic treatment during urologic procedures (e.g., TURP) [5,6]. In this respect, the ‘old’ antibiotic fosfomycin is of particular interest as many of the MDR bacteria remain susceptible to this antibiotic [7,8]. However, although recommended by the European Urology Association as an alternative, its use as prophylaxis during urologic procedures may be limited in many European countries, as it is currently not a registered indication of fosfomycin.

Ideally, the achieved prostatic tissue concentration of a prophylactic antibiotic is well above the minimal inhibitory concentration (MIC) of the uropathogen. However, with the present prophylactic dosing schemes, this is still unclear. Measuring antibiotic concentrations in prostate tissue is challenging as there is no standardized method. Options for measuring drug levels in tissue include the use of microdialysis, which measures unbound extracellular concentrations. An alternative is to grind up (washed) frozen tissue obtained in a chirurgical procedure (i.e., TURP). After the destruction of tissue cells, protein precipitation and centrifugation, drug levels can be measured in the supernatant.

In this study, we evaluated the concentrations of cefazolin, ciprofloxacin and fosfomycin in the human prostate in a cohort of men undergoing a TURP.

We compared prostate tissue concentrations to the serum concentrations and MIC (EUCAST breakpoints) of common uropathogens, to determine the appropriateness of the current presurgical prophylactic antibiotics and to gain supportive data about the suitability of fosfomycin for antibiotic prophylaxis in men undergoing urological procedures of the prostate.

## 2. Results

Seventy-five patients participated in the study, see Table 1 for baseline characteristics. There were 15 in the cefazolin group A, 14 in the ciprofloxacin group B1, 15 in the ciprofloxacin group B2, 15 in the fosfomycin group C1 and 16 in the fosfomycin group C2. All patients except one had more than 14 mL tissue chips removed and sufficient left-over prostate tissue available.

### 2.1. Group A: Intravenous Cefazolin 2 g Immediately before Surgery

The mean overall cefazolin serum concentration at the start of the surgery was 126.5 ± 19.1 µg/mL, at a mean time of 35 ± 4 min after administration. At the start of the surgery, a mean time of 19 ± 13 min after administration, a mean tissue concentration of 80.2 ± 38.6 µg/g was measured. At the end of the surgery, a mean time of 52 ± 15 min after administration, a mean tissue concentration of 40.2 ± 12.3 µg/g was measured. The mean tissue/serum ratio was 0.6 ± 0.3 at the start of the surgery. There was only one patient with a bodyweight of less than 80 kg, who received 1 g cefazolin. This resulted in a tissue concentration of 24 µg/g at the start of the surgery and 20 µg/g at the end of the surgery and a serum concentration of 86 mg/L (black dots in Figure 1). All serum and tissue concentrations were above the MIC of 4 µg/mL. The results are presented in Figure 1.

### 2.2. Group B1: Oral Ciprofloxacin 500 mg 1 h Prior to Surgery

The mean overall ciprofloxacin concentration at the start of the surgery was 2.3 ± 0.9 µg/mL, with a mean time of 70 ± 20 min after administration. At the start of the surgery, a mean time of 73 ± 24 min after administration, a mean tissue concentration of 4.9 ± 2.2 µg/g was measured. At the end of the surgery, a mean time of 105 ± 24 min after administration, a mean tissue concentration of 5.0 ± 2.2 µg/g was measured. The mean tissue/serum ratio was 2.2 ± 1.7 at the start of the surgery. There were three patients with undetectable serum and tissue concentrations, likely due to noncompliance. All other serum and tissue concentrations were above the MIC of 0.5 µg/mL. The results are presented in Figure 2.

### 2.3. Group B2: Oral Ciprofloxacin 500 mg 2 h Prior to Surgery

The mean overall ciprofloxacin concentration at the start of the surgery was 1.7 ± 0.3 µg/mL, at a mean time of 145 ± 42 min after administration. At the start of the surgery, a mean time of 239 ± 46 min after administration, a mean tissue concentration of 7.8 ± 4.2 µg/g was measured. At the end of the surgery, a mean time of 172 ± 52 min after administration, a mean tissue concentration of 5.6 ± 2.2 µg/g was measured. The mean tissue/serum ratio was 4.8 ± 2.5 at the start of the surgery. All serum and tissue concentrations were above the MIC of 0.5 µg/mL. The results are presented in Figure 3.

### 2.4. Group C1: Oral Fosfomycin 3 g 2 h Prior to Surgery

The mean fosfomycin serum concentration at the start of the surgery was 23.2 ± 8.6 µg/mL, at a mean time of 155 ± 62 min after administration. The mean fosfomycin concentration at the end of the surgery was 24.6 ± 8.0 µg/mL, at a mean time of 197 ± 77 min after administration. There was one missing sample. There were no results of the concentrations in prostate tissue due to matrix effects in the analysis and measurements below the LLOQ (3.75 µg/mL). The results are presented in Figure 4.

### 2.5. Group C2: Intravenous Fosfomycin 2 g Immediately before Surgery

The mean fosfomycin serum concentration at the start of the surgery was 108.0 ± 34.3 µg/mL, at a mean time of 16 ± 7 min after administration. One sample was drawn from the fosfomycin IV administration line and was rejected. One sample was missing and four samples were below the LLOQ. The mean fosfomycin serum concentration at the end of the surgery was 87.7 ± 16.1 µg/mL, at a mean time of 63 ± 17 min after administration. One sample was below the LLOQ. The fosfomycin concentrations in prostate tissue were below the LLOQ. The results are presented in Figure 5.

## 3. Discussion

In this study, we evaluated the concentration in serum and prostate tissue after a single dose of cefazolin, ciprofloxacin and fosfomycin in men undergoing TURP. These results show that after a single intravenous dose of cefazoline or an oral dose of ciprofloxacin prior to TURP, concentrations in serum and prostate tissue above the MIC of common uropathogens (*Enterobacterales*) were reached. These concentrations were also found in serum after a single intravenous dose of fosfomycin, but a single oral dose of fosfomycin did not lead to sufficiently high concentrations in serum.

All measured serum and tissue concentrations of cefazoline and ciprofloxacin were above MIC of common uropathogens (*Enterobacterales)*, both at the start and at the end of the procedure. For cefazolin, the mean tissue concentration/MIC ratio was 40 at the start of the procedure and 20 at the end of the procedure. For ciprofloxacin, at 1 h prior to surgery, the mean tissue concentration/MIC ratio was 5 at the start and at the end of the procedure. For ciprofloxacin, at 2 h prior to surgery, the mean tissue concentration/MIC ratio was 8 at the start of the procedure and 3 at the end of the procedure. Optimal tissue/MIC ratios have not been determined.

The mean tissue/serum ratio for ciprofloxacin is higher than that for cefazolin, indicating the better tissue penetration of ciprofloxacin. This may be explained by the higher protein binding of cefazolin (70–85%) compared to ciprofloxacin (20–30%) and therefore the lower free fraction of cefazolin. A weakness of this study is that serum and tissue samples were not drawn at the same time in all patients. Consequently, the ratio is only an indication of tissue penetration.

In the cefazolin group, only one participant had a bodyweight below 80 kg (70 kg) and therefore received 1 g cefazolin, which corresponded to 14 mg/kg. The average dose in all other participants was 23 mg/kg. The administration of 1 g resulted in lower serum and tissue concentrations, with a tissue concentration/MIC ratio of 10 at the start and the end of the procedure. This warrants a revision of the weight limit of 80 kg for the 2 g dose. It could be advisable to dose 2 g in case of body weight ≥ 65 kg.

To our knowledge, there are no reports of cefazolin concentrations in prostate tissue to date. There was a study by Ozturk showing a significant decrease in bacteremia incidence within 10 days after TURP after antibiotic prophylaxis with second- or third-generation cephalosporins (7%) compared to placebo (40%) [9]. Participants in this study received 1 g cefazolin 1 h prior to the surgery, regardless of body weight. Based on our results, it is likely that 1 g cefazolin (regardless of weight) will result in a tissue concentration > MIC, but given the short half-life there is a likelihood that the tissue concentration at the end of the procedure will not be sufficient. A single dose that is adjusted to bodyweight immediately prior to surgery leads to a tissue concentration that is effective during the entire procedure.

There were three patients with undetectable serum and tissue concentrations of ciprofloxacin. Likely the participants did not take the tablet even though it was provided by the nurse. Another option is an absorption problem due to interactions with other medication, food or delayed gastric emptying.

In an earlier study by Lugg, prostate tissue concentrations were measured after a single oral dose of 1000 mg extended-release ciprofloxacin. A mean tissue concentration of 4.75 ± 1.33 µg/g was found, with a mean serum concentration of 2.11 ± 1.00 µg/mL at 1 h after administration. The mean tissue/serum ratio at the start of the procedure was 1.9 [10]. This is consistent with our results, despite the use of a different dose and formulation.

Patients who have undergone a TURP have heterogeneous prostate tissue because of different underlying diseases. It is known that in patients with benign prostatic hypertrophy, the blood flow to the prostate is abnormal and the transition and the peripheral zone of the prostate have different degrees of vascularity. This results in heterogeneity of the prostate tissue [11]. The underlying disease and its severity are not included in the study; this may explain the large variation in the observed tissue concentrations.

As the tissue concentration/MIC ratio at the end of the surgery was higher, the administration of oral ciprofloxacin 2 h before surgery therefore seems more effective than administration at 1 h before surgery. Further research is needed to determine the clinical relevance of this.

A single dose of both oral and intravenous administration of fosfomycin led to serum concentrations above the MIC for uncomplicated urinary tract infections (8 µg/mL). Concentrations needed to treat other infections (MIC > 32 µg/mL) were only reached after a single dose of intravenous fosfomycin.

After intravenous administration, there were four patients with undetectable serum concentrations of fosfomycin at the start of the surgery. This can possibly be explained by accidental collection of the blood sample before the fosfomycin was administered.

Kuiper et al. reported a mean fosfomycin concentration of around 25 µg/mL in serum at 2 h after a single oral dose of 3 g [12]. This corresponds to the median concentration found in this study. The mean concentration after a single IV dose in our study is slightly lower than that found by Kuipers, but it is well above MIC (32 µg/mL) throughout the procedure. Zacarias et al. calculated a median maximum fosfomycin concentration of 23 µg/mL in serum after a single oral dose of 3 g [13].

We could not quantify fosfomycin in prostate tissue, which contradicts findings of earlier studies. Fosfomycin was measured in prostate tissue in a prospective study in healthy volunteers after a single oral dose of 3 g by Gardiner et al. [14]. They found a mean intraprostatic concentration of approximately 6.5 μg/g in the uninflamed prostate. Gardiner used a similar method for processing and analysis of the prostate tissue samples, but did not mention washing the samples before freezing. In another prospective study in patients with benign prostatic hyperplasia, parenteral fosfomycin achieved substantially higher intraprostatic concentrations of approximately 68.8 μg/g at 1 h following a single 4-gram dose [15].

In this study, prostate tissue was washed to remove all blood contamination and unbound cefazoline, ciprofloxacin or fosfomycin. After freezing and homogenization, intracellular antibiotic and tissue bound antibiotic were both measured.

Fosfomycin has no binding to protein and good tissue penetration is assumed. Due to the absence of protein binding, fosfomycin might have been washed away when washing the prostate tissue with sorbitol. That could explain why we did not measure bound or intracellular fosfomycin in prostate tissue in this study. Alternatively, the data provided by Gardiner could be too high, due to the attribution of blood-fosfomycin to the tissue level. Cefazoline has a protein binding of 70–85% and ciprofloxacin of 20–30%, resulting in greater drug binding to the prostate tissue so that it is not washed away. This may explain why we do find adequate concentrations cefazoline and ciprofloxacin in prostate tissue.

This study shows that after a single dose intravenous cefazoline or oral ciprofloxacin prior to TURP, concentrations in serum and prostate tissue above the MIC of common uropathogens (*Enterobacterales*) were reached. Both antibiotics seem potentially effective in preventing postsurgical infections and could be considered as presurgical prophylactic antibiotics in patients undergoing TURP. A single dose of oral and intravenous administration of fosfomycin both led to serum concentrations above the MIC for uncomplicated urinary tract infections (8 µg/mL). The MIC threshold needed to treat other infections, like complicated urinary tract infection (32 µg/mL) was only reached after a single dose of intravenous fosfomycin. We were unable to detect concentrations of fosfomycin in prostate tissue, possibly due to the methods used for processing and measuring fosfomycin in tissue. However, for prophylactic use, adequate serum concentrations or urine concentrations of antibiotics should be sufficient to prevent infections.

In view of the recent publications about the increased resistance and side effects of fluoroquinolones, alternative antibiotics should be considered [16,17]. Cai et al. suggest that fosfomycin would be a suitable alternative and this study confirms that [18].

Further clinical research is needed to determine the best method for measuring antibiotic concentrations in prostate tissue and the relationship between the concentration in prostate tissue and serum and the occurrence of postoperative prostatitis and bacteremia.

## 4. Materials and Methods

We prospectively recruited males who required TURP, at Haga Teaching Hospital, The Hague, The Netherlands, from September 2013 to July 2014 (Group A&B: cefazoline and ciprofloxacin prophylaxis) and from June 2016 to August 2017 (Group C: fosfomycin prophylaxis). Patients with severe renal impairment (defined as estimated glomerular filtration rate [eGFR] of <30 mL/minute/1.73 m^2^), a proven allergy to cefazolin, ciprofloxacin or fosfomycin or a urine culture with a resistant uropathogen for these antibiotics were excluded.

In the first part of this study, 15 patients received intravenous cefazolin, 1 g if bodyweight was <80 kg and 2 g if bodyweight was >80 kg, immediately prior to surgery (group A). Another 15 patients received 500 mg oral ciprofloxacin 1 h prior to surgery (group B1) and 15 patients received 500 mg oral ciprofloxacin 2 h prior to surgery (group B2). Both options were in accordance with the local protocol for presurgical antibiotic prophylaxis in Haga Teaching Hospital.

After this, we conducted a second part of this study with 15 patients receiving 3 g oral fosfomycin 2 h prior to surgery (group C1) and 15 patients receiving 2 g intravenous fosfomycin immediately prior to surgery (group C2). Fosfomycin was combined with standard antibiotic prophylaxis according to the local protocol.

At the start of the TURP procedure, a venous blood sample of 1 mL was drawn from an existing peripheral venous catheter. In group C, a second venous blood sample was drawn at the end of the TURP procedure. Time of sampling was recorded. All serum samples were stored at −80 °C until subsequent analysis.

After routine transurethral resection of the prostate, 10 mL of random tissue chips was selected for pathologic investigation. Leftover tissue was discarded. In this study, in the case that more than 14 mL of prostate chips was resected, the 2 mL of chips acquired first and the 2 mL that were acquired last were used. Cases with less than 14 mL tissue chips removed were excluded. Time of first removal of tissue and of last removal of tissue were accurately recorded.

### Analysis

Prostate samples were carefully washed with the irrigation liquid sorbitol 5% to remove all blood contamination and were weighed prior to being frozen at −80 °C until subsequent analysis. Sorbitol was used to facilitate sample freezing. Prior to analysis, the prostate tissue chips were homogenized with water using a IKA Ultra-Turrax^®^ Tube Drive Control with DT-20 tubes with rotor–stator element.

After protein precipitation, all samples were centrifuged for 10 min at 13,000 rpm and supernatant was analyzed using LC-MS/MS.

The cefazolin and ciprofloxacin concentrations in serum and prostate tissue supernatant samples were determined by a validated liquid chromatography–tandem mass spectrometry (LC-MS/MS) analysis method. Standard curves were developed for serum as well as prostate levels and an internal standard (haloperidol 10 µg/L and ciprofloxacin-D8 HCl 350 µg/L in 85% ACN/15% MeOH) was used.

Fosfomycin concentrations in serum were analyzed using a validated LC-MS/MS method [19]. The upper and lower limits of quantification were 375 and 3.75 mg/L, respectively. Results above the upper limit of quantification were diluted and re-analyzed. Prostate tissue was also analyzed with this method, although it was not validated for this matrix.

## Figures and Tables

**Figure 1 antibiotics-12-00022-f001:**
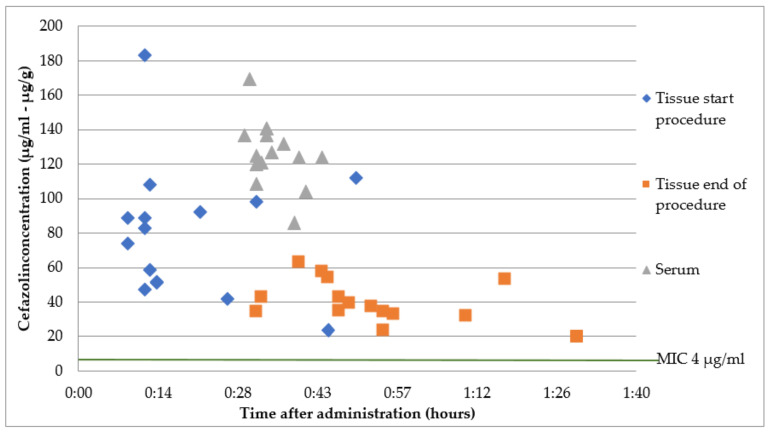
Cefazolin concentrations in serum and prostate tissue after a single intravenous dose immediately prior to surgery. *Enterobacterales* are considered susceptible to cefazolin when MIC < 4 µg/mL [7].

**Figure 2 antibiotics-12-00022-f002:**
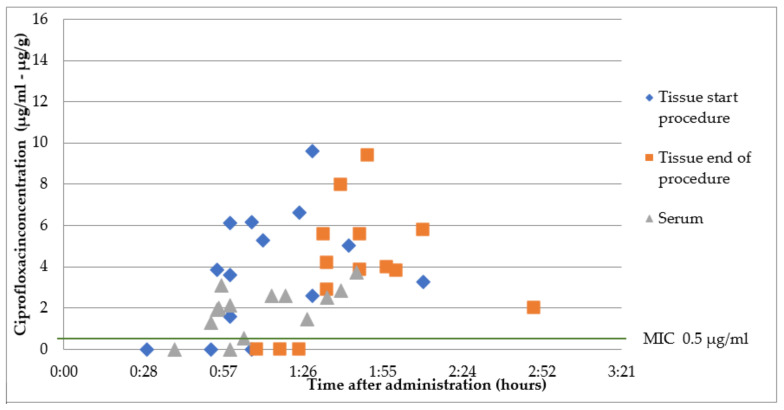
Ciprofloxacin concentrations in serum and prostate tissue after a single oral dose 1 h prior to surgery. *Enterobacterales* are considered susceptible to fluoroquinolones when MIC < 0.5 µg/mL [7].

**Figure 3 antibiotics-12-00022-f003:**
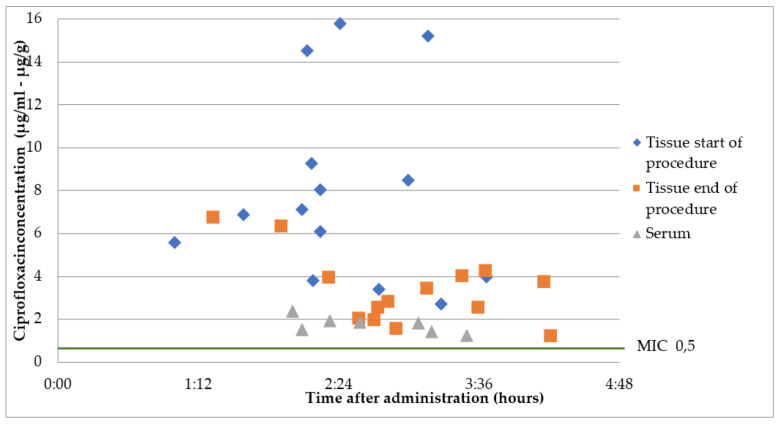
Ciprofloxacin concentrations in serum and prostate tissue after a single oral dose 2 h prior to surgery. *Enterobacterales* are considered susceptible to fluoroquinolones when MIC < 0.5 µg/mL [7].

**Figure 4 antibiotics-12-00022-f004:**
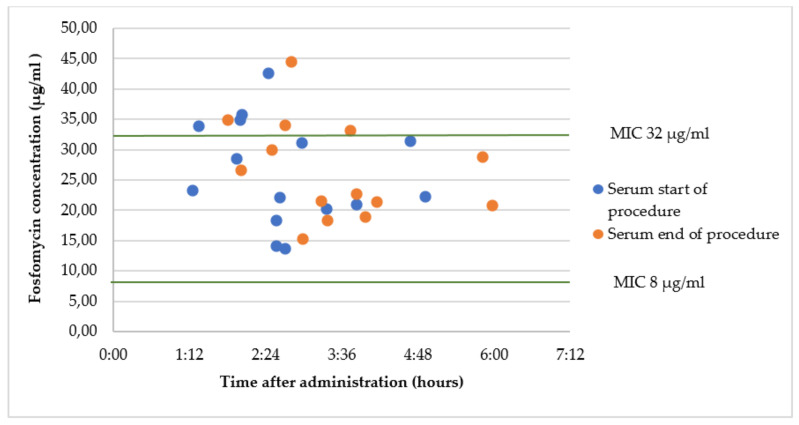
Fosfomycin concentrations in serum after a single oral dose 2 h prior to surgery. *E. coli* is considered susceptible to oral fosfomycin for treatment of urinary tract infection when MIC < 8 µg/mL and *Enterobacterales* are considered susceptible to oral fosfomycin when MIC < 32 µg/mL [7].

**Figure 5 antibiotics-12-00022-f005:**
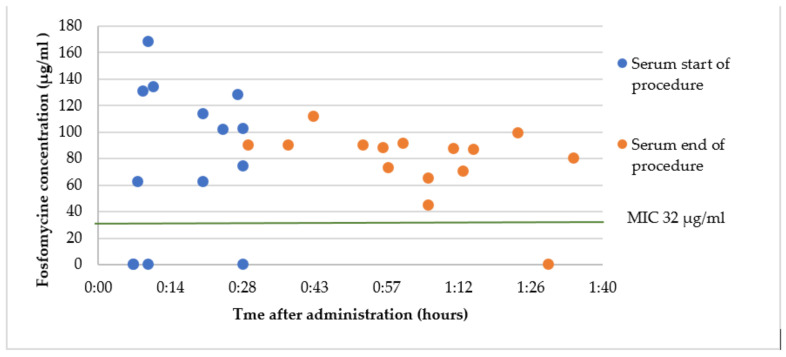
Fosfomycin concentrations in serum after a single intravenous dose immediately prior to surgery. *Enterobacterales* are considered susceptible to intravenous fosfomycin when MIC < 32 µg/mL [7].

**Table 1 antibiotics-12-00022-t001:** Baseline characteristics.

	Group A (n = 15)	Group B1 (n = 14)	Group B2 (n = 16)	Group C1 (n = 15)	Group C2 (n = 15)
Age, median, years (range)	69 (55–76)	76 (54–93)	73 (58–88)	67 (61–84)	72 (57–88)
Weight, mean, kg (SD)	88 (±8)	84 (±16)	80 (±11)	80 (±16)	79 (±22)
eGFR, mL/min/1.73 m2, MDRD, mean (SD)	82 (±18)	66 (±17)	76 (±13)	92 (±42)	87 (±24)

Group A, cefazoline 2 gr before surgery: group B1, oral ciprofloxacin 500 mg 1 h prior to surgery; group B2, oral ciprofloxacin 500 mg 2 h prior to surgery; group C1, oral fosfomycin 3 g 2 h prior to surgery; group C2, intravenous fosfomycin 2 g immediately before surgery. eGFR: estimated glomerular filtration rate. MDRD: Modification of Diet in Renal Diseases.

## Data Availability

Not applicable.

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
