# Peer review of "Serum and Prostatic Tissue Concentrations of Cefazolin, Ciprofloxacin and Fosfomycin after Prophylactic Use for Transurethral Resection of the Prostate"

_antibiotics, 2022, doi:10.3390/antibiotics12010022_

Round 1
Reviewer 1 Report
The Authors presented an original study of serum and prostate tissue concentrations of different antibiotics used as prophylaxis in men undergoing TURP. The paper is generally well written and the topic is of clinical relevance. Here are a few tips:
Abstract:
- “men” instead of “man” seems more appropriate.
- The authors should clarify also fosfomycin concentrations in prostate tissue (as stated for cefazoline and ciprofloxacin)
- The MICs you refer to are EUCAST breakpoints and this should be clarified (they are not MICs based on local epidemiology or reported in previous studies)
Manuscript:
- Line 49: replace “particularly” with “particular”
- Line 63, 67, 154: replace “man” with “men”
- Table 1: weight group C2: I think 7 kg is incorrect
- Line 80: remove “at”
- Line 88: The MIC you refer to is EUCAST breakpoint. This should be clarified, as indicated in the abstract. This comment also refers to other lines in the text. The Authors may consider clarify in the Materials and Methods section that the MICs they refer to in the text are EUCAST breakpoints.
- Figures 1-5: “time after administration”, it should be clarified the unit of measurement (hours).
- Lines 84, 102, 117: unit of measurement is missing
- Lines 134: E. coli should be written in italics
- Enterobacterales should be written in italics (several lines in the manuscript)
- Lines 156-157: “concentrations in serum and prostate tissue, are well above the MIC of common uropathogens (Enterobacterales) were reached” does not make sense: “are well above” or “were reached” should be removed.
- Consider summing up strengths and limitations of your study in a dedicated section instead of stating them in various parts of the Discussion.
- Was Ethical Approval obtained?
Author Response
Abstract:
- “men” instead of “man” seems more appropriate. - adjusted
- The authors should clarify also fosfomycin concentrations in prostate tissue (as stated for cefazoline and ciprofloxacin) - Comment added to the abstract
- The MICs you refer to are EUCAST breakpoints and this should be clarified (they are not MICs based on local epidemiology or reported in previous studies) - explanation added to the abstract and in the introduction
Manuscript:
- Line 49: replace “particularly” with “particular” - adjusted
- Line 63, 67, 154: replace “man” with “men” – adjusted
- Table 1: weight group C2: I think 7 kg is incorrect – adjusted to 79 kg
- Line 80: remove “at” à adjusted
- Line 88: The MIC you refer to is EUCAST breakpoint. This should be clarified, as indicated in the abstract. This comment also refers to other lines in the text. The Authors may consider clarify in the Materials and Methods section that the MICs they refer to in the text are EUCAST breakpoints. - explanation added to the abstract and in the introduction
- Figures 1-5: “time after administration”, it should be clarified the unit of measurement (hours). - adjusted
- Lines 84, 102, 117: unit of measurement is missing - adjusted
- Lines 134: E. coli should be written in italics - adjusted
- Enterobacterales should be written in italics (several lines in the manuscript) - adjusted
- Lines 156-157: “concentrations in serum and prostate tissue, are well above the MIC of common uropathogens (Enterobacterales) were reached” does not make sense: “are well above” or “were reached” should be removed. – adjusted
- Consider summing up strengths and limitations of your study in a dedicated section instead of stating them in various parts of the Discussion. - the discussion has been modified to better reflect strengths and weaknesses.
- Was Ethical Approval obtained? à yes, see lines 321-323
Reviewer 2 Report
Authors should be congratulated for their work. The prophylaxis for the urological procedure is still a hot topic in the medical field. The major concern of the medical assumption is related to the multidrug-resistance pathogens spreading. The paper is well written but several observations are noteworthy:
- first, how the processing of the turp chips could affect the antibiotic concentrations studied?
- Are data available on the population studied? How is their hepatic function? How is big the prostate? How the prostate volume influences the MIC level?
- Are data available on the number of patients that develop a urinary tract infection after treatment? What are the pathogens involved? Are MDR bacteria?
Author Response
- first, how the processing of the turp chips could affect the antibiotic concentrations studied? - see lines 236-244
- Are data available on the population studied? How is their hepatic function? How is big the prostate? How the prostate volume influences the MIC level? - this data was not included in the protocol and was not collected in the study.
- Are data available on the number of patients that develop a urinary tract infection after treatment? What are the pathogens involved? Are MDR bacteria? - this data was not included in the protocol and was not collected in the study. This will be taken into account in a possible follow-up study
Round 2
Reviewer 2 Report
The authors improved their manuscript, clarifying the weak points.